# Non-Mutational Key Features in the Biology of Thymomas

**DOI:** 10.3390/cancers16050942

**Published:** 2024-02-26

**Authors:** Stefan Küffer, Denise Müller, Alexander Marx, Philipp Ströbel

**Affiliations:** Institute of Pathology, University Medical Center Göttingen, University of Göttingen, 37075 Göttingen, Germany; stefan.kueffer@med.uni-goettingen.de (S.K.); denise.mueller@med.uni-goettingen.de (D.M.); alexander.marx@med.uni-goettingen.de (A.M.)

**Keywords:** thymus, thymoma, oncogenic driver, mutations, chromosomal aberrations, RNA, methylation, gene expression, apoptosis, proteomics, metabolomics

## Abstract

**Simple Summary:**

Organotypic features such as intratumoral thymopoiesis make thymomas (THs) unique tumors. Their remarkable histological heterogeneity and low mutational burden have largely precluded personalized therapies. Although most thymomas do not harbor known oncogenic driver mutations, common non-mutational key features such as chromosomal, epigenetic, and metabolic alterations are emerging that converge into a limited number of critical cellular pathways that may provide opportunities for therapeutic interventions. In this review, we have attempted to integrate the existing knowledge of TH biology into a broader picture and highlight opportunities for targeted treatment options in these non-mutated tumors.

**Abstract:**

Thymomas (THs) are a unique group of heterogeneous tumors of the thymic epithelium. In particular, the subtypes B2 and B3 tend to be aggressive and metastatic. Radical tumor resection remains the only curative option for localized tumors, while more advanced THs require multimodal treatment. Deep sequencing analyses have failed to identify known oncogenic driver mutations in TH, with the notable exception of the *GTF2I* mutation, which occurs predominantly in type A and AB THs. However, there are multiple alternative non-mutational mechanisms (e.g., perturbed thymic developmental programs, metabolism, non-coding RNA networks) that control cellular behavior and tumorigenesis through the deregulation of critical molecular pathways. Here, we attempted to show how the results of studies investigating such alternative mechanisms could be integrated into a current model of TH biology. This model could be used to focus ongoing research and therapeutic strategies.

## 1. Introduction

Thymomas (THs) are rare epithelial tumors of the anterior mediastinum. Their most common characteristic feature is the variable preservation of important “organotypic” features of the normal thymus, in particular, the ability to promote the maturation of T cells from immature precursors [1]. Based on the morphology of the neoplastic epithelial cells and the relative proportion of immature T cells, the World Health Organization (WHO) recognizes five main TH subtypes (A, AB, B1, B2, and B3) with different epidemiologic, histologic, and clinical characteristics such as their propensity to present in advanced stages or to metastasize [2] (Figure 1). 

Probably due to their ability to promote and export tumor-derived T cells, approximately 40% of THs are associated with paraneoplastic autoimmune diseases, the most common of which is myasthenia gravis (MG) [3]. These organotypic features distinguish THs from the more aggressive thymic carcinomas (TCs), which have lost the ability to promote T cell maturation and are therefore not associated with MG. Another highly characteristic and fundamental difference between THs and TCs (thymic epithelial tumors, TETs) is their pattern of progression, with prolonged confinement to the thoracic cavity and the virtual absence of lymph node metastasis in THs. Molecular data from The Cancer Genome Atlas (TCGA) program have confirmed that THs and TCs are distinct entities [4]. Among THs, the data suggest that type A and AB are distantly related neoplasms. In contrast, type B thymomas form a different cluster that appears to represent a spectrum rather than separate entities, an observation supported by the frequent finding of more than one type B component in a given tumor. These observations are important since they point to different molecular routes in the pathogenesis of the four main thymic epithelial tumor (TET) groups. While many THs are slow-growing and relatively indolent neoplasms, type B2 and B3 THs are more aggressive and may present in advanced clinical stages with the infiltration of adjacent anatomic structures or intrathoracic metastases. Advanced THs are treated by a multimodal approach with systemic chemotherapy, surgery, and radiotherapy [5]. The TCGA study has confirmed the notion that THs are among the adult tumors with the lowest mutational burden and very few recurrent gene mutations, with the notable exception of *GTF2I* mutations, which are preferentially found in type A and AB thymomas [4,6]. This observation may explain why, thus far, there has been little benefit from molecularly targeted approaches in patients with THs, even compared to TCs, where there are now a couple of therapeutic options available. However, while this may have come as a disappointment to many, a vast body of literature over the decades has helped to pave the way toward a better understanding of the driving factors behind the biology of THs. Based on current knowledge, it seems plausible to assume that TETs develop along (at least) three different molecular routes: (A) Type A and AB thymomas with few chromosomal alterations but with a highly specific and recurrent *GTF2I* mutation as a fundamental driver, (B) type B thymomas with increasing numbers of chromosomal alterations, but so far no known driver mutations, suggesting alternative, non-mutational mechanisms determining their behavior, and (C) thymic carcinomas with multiple chromosomal alterations and several recurrent gene mutations (reviewed elsewhere in this Cancers Special Issue). Through multiple subtype-specific mechanisms (Figure 2), these alternative molecular routes appear to converge in at least seven critical cellular pathways that can be considered the hallmarks of TH biology: (1) Wnt signaling, (2) altered receptor tyrosine kinase (RTK) signaling, (3) RAS/MAPK/ERK signaling, (4) PI3K/AKT/mTOR signaling, (5) cell cycle regulation, (6) cellular senescence and apoptosis resistance, and (7) epigenetic regulation. This list is probably incomplete, and other mechanisms may emerge as the modern bioinformatic analysis of multi-omics helps better manage the complexity of the available data. In this comprehensive narrative review of the scientific English literature listed in PubMed from 1990 to 2024, we have attempted to summarize and highlight the current knowledge about the non-mutational drivers of THs. The fact that thymomas are not only malignant neoplasms but are also uniquely capable of distorting the immune system adds another layer of complexity that is intrinsic to the biology of these tumors and cannot be ignored but is beyond the scope of this review. Immunotherapy in THs is described elsewhere in this Special Issue of Cancers. 

## 2. Disturbed Thymic Developmental Programs in Thymomas

Most THs contain subtype-specific mixtures of functional, albeit defective, cells with cortical and medullary differentiation, suggesting their origin from bipotent thymic progenitors [7]. The absence of the medullary autoimmune regulator gene (AIRE) in 95% of THs and recent studies in mice with mutations in *GTF2I* have provided experimental support for this hypothesis [8,9]. Therefore, it is to be expected that many of the programs and factors that govern the development and maintenance of the normal thymus are also important in the pathogenesis of THs. Cortical and medullary epithelial cells (cTECs and mTECs) of the normal thymus develop from a single bi-potent endodermal progenitor cell [10,11]. Recent single-cell analyses have revealed the vast complexity of thymic epithelial cells (TECs) with at least nine distinct TEC subsets including two cTEC and three mTEC clusters, one immature or uncommitted TEC cluster with canonical TEC identity genes (*FOXN1*, *PAX9*, *SIX1*) as well as three more clusters with neuroendocrine (*BEX1*, *NEUROD1*), muscle-like myoid (*MYOD1*, *DES*), and myelin-epithelial (*SOX10*, *MPZ*) cell features [12]. Major signaling pathways that play a critical role in TEC development and function include WNT, BMP, transforming growth factor beta (TGF beta), insulin-like growth factor (IGF), and fibroblast growth factor (FGF) signaling [13,14,15,16].

WNT signaling is the best studied in THs of the factors listed above. WNT signaling plays a key role in the development and maintenance of the thymus, and the decreased expression of WNT proteins or increased levels of WNT inhibitors is associated with TEC senescence and thymic involution in mice and humans [17,18,19]. WNT4 is secreted from normal thymic epithelial cells and activates a signaling network via G-protein-dependent Frizzled receptors in an autocrine manner [20]. This autocrine loop is also active in THs, especially in the more aggressive subtypes [21]. Importantly, WNT signaling in THs appears to be non-canonical (i.e., β-catenin-independent) through the so-called PCP/JNK pathway [22] and may be supported by AKT and NF-kB signaling [21]. Wnt signaling also activates FOXN1 [14], a critical transcription factor that is deficient in mice and humans with the nude phenotype [23] and is essential for the differentiation and maturation of all TEC [24]. FOXN1 promotes the proliferation of thymic epithelial progenitor cells and mTECs in the normal thymus [25] and in thymoma cells [26,27]. The FOXN1 protein is expressed in most thymomas but partially lost in TCs [28]. 

GTF2I is another transcription factor with a role in thymic development that is preferentially mutated in type A and AB thymomas [6,29]. However, a recent study using microdissected tissues reported high mutation rates of 42% also in type B thymomas [30]. In endogenous mouse models with the specific expression of mutated *GTF2I* in FOXN1+ thymic epithelial cells, the mutation causes an incomplete block of TEC differentiation with the accumulation of immature TECs and reduced thymopoietic activity [8] and leads to the formation of thymomas resembling WHO type B1 and B2 THs in aged animals [31]. The mutation consistently confers increased resistance to apoptosis, tolerance to DNA damage, and alterations in cell cycle regulation with enrichment of the MYC, E2F, and G2M checkpoint target genes [8,31,32]. In addition, the mutation is associated with a growth advantage of the transformed cells through the activation of glycolysis and altered lipid biosynthesis [32]. THs with *GTF2I* mutations showed increased expression of genes related to cell morphogenesis, receptor tyrosine kinase signaling, retinoic acid receptors, neuronal processes, and WNT and SHH signaling. Downregulated pathways included apoptosis, cell cycle, DNA damage response, hormone receptor signaling, RAS/MAPK, and TSC/mTOR [4]. GTF2I is a critical factor in the pathogenesis of the neurodevelopmental 7q11.23 microduplication autism spectrum disorder [33], where it is responsible for transcriptional dysregulation in early pluripotent neuronal stem cells by repressing critical neuronal genes through the recruitment of lysine demethylase 1 (LSD1) [34]. Importantly, it has been shown in patient-derived neuronal cells that histone deacetylase inhibitors (HDACi) can reduce abnormal expression levels of GTF2I [33]. This therapeutic option could also be further pursued in thymoma patients with *GTF2I* mutations.

## 3. Functional Implications of Chromosomal Alterations in Thymomas

Chromosomal aberrations are important because they were among the first molecular findings in THs [35] and are highly characteristic and reproducible, although their functional implications are still poorly understood. Their frequency increases from type A to B3 THs and correlates with the increasing aggressiveness of these subtypes [4,36]. Large-scale, whole-, and arm-level somatic copy number alterations (sCNAs) occur predominantly in tumors without gene mutations [4]. 

Type A showed very few arm-level copy number alterations [36]. The most common arm-level chromosomal alterations in thymomas (and in thymic squamous cell carcinomas) included losses on chromosome 6p25.1-p24.3, 6q25.3, 9p12.3, 13q and 16q, 22p12.1-p12.2 and gains on chromosome 1q31.1, 1q43, 3p12.2, 8q23.1, 9p24.3, 10q21.1, 12p12.3, 14q12, and Xq23 andXq25 [4,36]. Type B3 THs and TCs showed overlapping chromosomal alterations with frequent arm-level CN gains of 1q and losses of chromosomes 6 and 13q. This finding starkly contrasts an integrative unsupervised clustering of five data platforms that suggested that TCs are profoundly different from thymomas [4]. Petrini et al., were the first to correlate these copy number alterations with known cancer-related genes [36]. They identified *BCL2* as a likely target of copy number gain at chr. 18q and *CDKN2A* as a likely target of loss at chr. 9p21. *CDKN2A* is a tumor suppressor gene that encodes for the two tumor suppressor proteins, p14 and p16 [37]. *CDKN2A* mutations are found in approximately 6% of THs compared to 40% in TCs [38,39]. p16 exerts its functions by inducing senescence in early transformed cells or by inducing apoptosis through TP53, the inhibition of the cyclin-dependent kinases CDK4 and CDK6, and the restoration of the retinoblastoma protein during the G1/S cell cycle phase [40,41]. Importantly, the loss of p16 expression is not exclusively dependent on the CN loss of *CDKN2A* but can also occur as a consequence of p16 promoter methylation [36,42,43] or miR-24 deregulation [44] and its loss correlates with tumor aggressiveness [36,45]. Together, these data arguably make *CDKN2A/p16* one of the most relevant altered genes in thymic epithelial tumors. Another highly important finding of the study by Petrini et al., was the identification of the profound deregulation of BCL2 family genes in THs, namely CN gains of *BCL2* in tumors with chromosomal gains at chr. 18q or focal gene amplifications of *BCL2* [36]. Frequent overexpression of the BCL2 protein in TETs had already been previously described [46,47,48,49]. Moreover, the same study also revealed CN gains of *MCL-1* [50] and *BCL-xL* in a substantial number of cases [36]. The fact that anti-apoptosis plays a central role in TET biology is further supported by previous observations describing the overexpression of the anti-apoptotic protein BIRC3 in TCs, the downregulation of the pro-apoptotic protein NOXA in type B3 THs [51], and the overexpression of the anti-apoptotic cellular FLICE-like inhibitory protein cFLIP [52]. Furthermore, functional studies using BH3 profiling in cell lines and tissue samples showed that B2 and B3 THs are exquisitely dependent on pro-survival factors such as MCL-1 and BCL-xL [36,53]. These observations suggest that TETs are promising candidates for clinical trials (e.g., with inhibitors of MCL-1 and BCL-xL).

The TCGA dataset [4] has extended the correlations between chromosomal alterations and potential genes of interest in commonly deleted or amplified chromosomal regions that await further experimental validation: *PTPRG*, *FOXP1*, and *RYBP* (3p13), *CDYL* (6p25.1–24.3), *ARID1B*, *ZDHHC14* (6q25.3), NF2, CHEK2 (22p12.1-p12.2), *PTGS2* (1q31.1), *ACTN2* (1q43), *PTPRD* (9p24.3), *PIK3C2G* (12p12.3), and *FOXG1* (14q12). 

A recent publication suggests that the frequency of recurrent gene fusions in THs may have been underestimated. Based on RNA seq data, Ji et al., identified gene fusions in 36% (9/25 samples) including one sample with multiple gene fusions [54]. Of these, five fusions were unique to THs and included the previously reported *KMT2A-MAML2* gene fusion [55] in 36% of cases as well as *HADHB-REEP1*, *COQ3-CGA*, *MCM4-SNTB1*, and *IFT140-ACTN4*. The *KMT2A* gene encodes histone-lysine N-methyltransferase 2A, also known as acute lymphoblastic leukemia 1 (ALL-1), myeloid/lymphoid, or mixed-lineage leukemia 1 (MLL1). MLL1 is a positive global regulator of gene transcription with an essential role in regulating *Hox* genes [56]. MAML2 is a member of a family of NOTCH signaling coactivators. The *KMT2A-MAML2* fusion gene has been shown to be oncogenic [57]. It disrupts NOTCH signaling and suppresses *HES1* promoter activation [58]. A single published case of a thymoma with a neurotrophic tyrosine receptor kinase (*NTRK*) gene fusion showed a significant response to entrectinib [59].

## 4. Altered DNA Methylation and Acetylation in Thymomas

DNA methyltransferase (DNMT) enzymes facilitate DNA methylation and significantly affect gene expression. DNMTs establish and maintain methylation patterns by adding methyl groups to cytosine residues within CpG dinucleotides. Alterations in CpG methylation contribute to cancer by causing the hypermethylation of gene promoters, leading to transcriptional silencing of tumor suppressor genes and hypomethylation of intergenic and intronic sequences, inducing chromosomal instability in proto-oncogene regions. In mammalian genomes, DNMT3a and DNMT3b are responsible for de novo methylation, while DNMT1 is responsible for maintaining methylation patterns. Increased DNMT activity in cancer cells is associated with tumor aggressiveness and poor patient prognosis [60]. Numerous published studies, summarized in recent reviews [61,62], have addressed the epigenetics of TETs. Most of the available literature has focused on the potential use of DNA methylation markers to discriminate between different histologic subtypes or between THs with and without myasthenia gravis or as a prognostic indicator. Studies analyzing global methylation patterns in TETs have found substantial hyper- or hypomethylation of CpG sites with significant correlations with the TH histologic subtype [4,63,64,65], *GTF2I* or *HRAS* mutation status, and mRNA and miRNA expression patterns [4]. Previous studies using a candidate gene approach revealed the hypomethylation of several tumor suppressor genes (e.g., *APC1A*, *CDKN2A*, *E-cad*, *FHIT*, *hMLH1*, *MGMT*, *RARβ*, *RASSF1A*) or hypermethylation of *DNMT1*, *DNMT3a*, and *DNMT3b* [66] in early stage THs compared to more aggressive or late stage THs and TCs (reviewed in [61,62]). The specific silencing of these genes with a role in regulating cell cycle progression and DNA repair suggests an essential step during TH progression.

Acetylation is a covalent histone modification that occurs at evolutionarily conserved lysine residues of nucleosome complexes. Acetyl groups added to histones loosen their chemical interactions with DNA, promoting the decompaction of tightly packed chromatin. This process enhances accessibility to transcriptional machinery components, establishing permissive chromatin states for gene expression [67]. Conversely, the removal of acetyl groups from histones by histone deacetylases (HDACs) generally results in the transcriptional repression and silencing of tumor suppressor genes. Overexpression of class I (HDAC1-3 and 8) and class II (HDAC4-7 and 9-10) HDACs is associated with poor prognosis in THs [68]. The HDAC inhibitor belinostat has shown activity in THs in vitro [69] and clinical trials [70,71].

## 5. Altered Gene Expression and Non-Coding RNA Networks in Thymomas

Many studies have analyzed mRNA expression profiles in TETs [4,13,72,73,74,75,76,77,78,79,80,81]. However, the results are difficult to compare due to differences in the study focus, methodology, number of samples, controls (e.g., TH vs. normal thymus, TH vs. TC, TH with and without myasthenia gravis). Recent studies have shown a high concordance with the main TH subgroups [4,13]. THs with *GTF2I* mutations (type A and AB thymomas) were enriched for genes related to human embryonic stem cell pluripotency and Wnt/β-catenin signaling [82], lymphocyte-rich type B1 and B2 thymomas were enriched for genes related to T-cell receptor signaling, CTLA4 signaling, and ICOS/ICOSL signaling, whereas type B3 thymomas overexpressed genes related to G protein-coupled receptor signaling, Toll-like receptor signaling regulation of epithelial–mesenchymal transition (EMT) as well as NF-κB, EGF, FAK, and telomerase signaling. Differentially regulated pathways that were identified in more than one study included cell cycle regulation [4,72,79,81], MAPK/ERK signaling [4,78], hormone signaling [4,72], Wnt signaling [78,82], and apoptosis [4,81]. In a study by Badve et al., the top pathways associated with metastasis were amino acid metabolism, steroid and glycosphingolipid biosynthesis, cell cycle checkpoint proteins, and Notch signaling [72].

Non-coding RNAs (ncRNAs) play critical roles in various biological processes, diseases and tumors, and many published studies have shown that their expression patterns can be used for the accurate classification and prognostic prediction of THs. The role of ncRNAs in TETs has been addressed in several recent excellent reviews [61,62,83] and will only be briefly highlighted here. ncRNAs are produced by transcription from various genomic regions and undergo post-transcriptional maturation and modification [84]. They can be classified into small non-coding RNAs (miRNAs and siRNAs) and long non-coding RNAs (lncRNAs), which can be further subdivided into circular and linear forms [85]. LncRNAs have emerged as important modulators of gene expression, affecting networks in different human cancers including TETs [86]. Acting at different levels of control in gene regulation, lncRNAs affect pathways related to cell fate determination in normal and pathological conditions. Circular lncRNAs (circRNAs), characterized by covalently closed structures, exhibit stability and diverse functions such as serving as microRNA sponges, modulating RNA-binding proteins, or acting as protein scaffolds [87]. Studies have shown differential expression of circRNAs in THs [88]. Studies of lncRNA profiles in TH tissues suggest that specific lncRNA expression alterations may be associated with overall or disease-free survival in THs. Su et al., identified four lncRNAs (ADAMTS9-AS1, HSD52, LINC00968, and LINC01697) that were significantly associated with recurrence-free survival (RFS) in TETs [89]. These lncRNAs stratified patients into high- or low-risk groups, outperforming traditional prognostic models. In a separate study of 25 TH patients, ADAMTS9-AS1, which was associated with RFS, showed altered expression with five other lncRNAs affecting various cancer types and biological processes [54]. In particular, ADAMTS9-AS1 and LINC00324 expression correlated with patient prognosis. These lncRNAs were associated with miRNA cluster dysregulation and target gene alterations in tumorigenic signaling pathways such as PI3K/Akt, FoxO, HIF-1, and Notch, supporting their roles in TH progression. Using data mining approaches, two groups found that upregulation of AFAP1-AS1 and downregulation of LINC00324 were associated with RFS in TET patients [54,90]. Ji et al., performed a functional annotation of the top 10 lncRNAs with altered expression in THs and identified many predicted target genes [54] that merit further functional investigation. In addition, the lncRNA RP11-424C20.2 regulates *UHRF1* expression by sponging miR-378a-3p, which influences thymoma prognosis through interactions with infiltrating immune cells and modulation of the tumor immune microenvironment [91,92]. Among small ncRNAs, microRNAs (miRNAs) are prominent in post-transcriptional regulation by binding to specific mRNA targets and inducing degradation or translational inhibition [93]. miRNAs can also repress key enzymes that drive epigenetic remodeling [94]. Radovich et al., identified selective overexpression of a large microRNA cluster on chr19q13.42 (C19MC) exclusively in type A and AB TH, which was also confirmed by TCGA data [4]. Overexpression of this microRNA cluster activates the PI3K/AKT/mTOR pathway, and PI3K/AKT/mTOR inhibitors reduced cell viability in a type AB TH cell line [77]. Another miRNA cluster on chr14q32 (C14MC) is expressed in THs and downregulated in TCs [95]. Integrating a large amount of TH data into a lncRNA-mRNA-miRNA regulatory network, Ji et al., predicted that the overexpression of miRNA clusters activates the PI3K-Akt/FoxO/HIF-1/Rap-1 signaling pathway [54]. Data from the TCGA study [96] showed that high expression of LOXL1-AS1 and low expression of miR-525-5p were associated with poor prognosis in TETs. LOXL1-AS1 acts as a sponge for miR-525-5p, promoting HSPA9 expression and thereby enhancing the growth and invasion of TET cells while inhibiting apoptosis [96]. MiR-525-5p acts as a tumor suppressor by repressing HSPA9, which is upregulated in TETs and correlates with poor patient survival. Another network involves LINC00174, miR-145-5p, and target genes in thymic tumorigenesis [97]. Upregulation of LINC00174 is negatively associated with miR-145-5p [98] and suggests an oncogenic role in TETs, affecting cell growth, migration, and lipid metabolism. The lncRNA MALAT1, acting as a miR-145-5p sponge, contributes to thymic cancer development, with its downregulation leading to decreased proliferation and increased apoptosis. The combination of MALAT1 silencing and miR-145-5p overexpression has a synergistic effect [99]. Iaiza et al., found that methylation and delocalization of *MALAT1* through the methyltransferase METTL3 induces *c-MYC* expression in aggressive THs and TCs. Silencing of METTL3 combined with cisplatin or c-MYC inhibitors promotes apoptosis in TC cells [100]. 

## 6. Metabolic Reprogramming in Thymomas

The metabolome of THs and TCs is an evolving field and remains understudied. Circumstantial evidence that THs, especially the aggressive types, show profound changes in their metabolism comes from nuclear medicine and positron emission tomography (PET) studies using fluorodeoxyglucose. It has been shown that high preoperative glucose uptake [101,102] and high serum levels of lactate dehydrogenase (LDH) [103] can be used to predict histologic subtypes and prognosis in thymoma patients. These observations were consistent with a metabolic study in THs showing high lactic acid and glutamine levels and activation of the proline/arginine, glycolysis, and glutathione pathways, suggesting increased glycolysis and glutaminolysis [104]. Two other studies concluded that THs contain two main metabolic subgroups [105,106]. Zhang et al., identified alterations in neutral lipid biosynthesis and phospholipid metabolism, particularly the lacto and neolacto series in glycosphingolipid biosynthesis, as a central metabolic pathway associated with TH progression [105]. They identified two key enzymes of this pathway, B3GNT5, and ST3GAL6, which showed prognostic correlation with the overall survival of TH patients. Glycosphingolipids are a family of essential cell membrane molecules involved in cell–cell recognition and signal transduction [107]. Changes in their glycosylation are associated with stem cell differentiation and various cancer-related processes such as cell proliferation and metastasis [108]. Overexpression of B3GNT5, an essential enzyme in producing lactate and lactate-series glycosphingolipids, was associated with poor outcomes in TH patients. The other enzyme, ST3GAL6, is involved in synthesizing glycolipid substrates and is altered in various cancers. Consistent with findings in lung cancer [109], where downregulation of ST3GAL6 activates EGFR/MAPK signaling and stimulates the expression of matrix metalloproteinase 2 and 9, low levels predicted poor prognosis in TH patients. Tang et al. [106] identified two distinct metabolic patterns in THs, with different metabolic scores correlating with clinical outcomes. High metabolic scores were associated with worse survival and immunosuppressive status. The study also described differential activation of glycosphingolipid metabolism and arginine biosynthesis. Interestingly, a recent proteomic study has described overexpression of the rate-limiting enzyme for arginine biosynthesis, argininosuccinate synthase 1 (ASS1), in type B2 and B3 compared to type A, AB, and B1 THs [110]. The upregulation of ASS1 in malignant THs contrasts with many other tumors (e.g., hepatocellular carcinoma or mesothelioma) that downregulate ASS1 and consequently become dependent on extracellular arginine [111]. In addition, asparagine synthetase (ASNS), an enzyme involved in amino acid metabolism that requires glutamine to convert aspartic acid to asparagine, was upregulated in THs and correlated with reduced patient survival [106]. 

## 7. Proteomics and Altered Tyrosine Kinase Signaling

The proteome of THs is still a relatively unexplored area of research. A number of publications have analyzed the thymoma proteome in an unbiased manner using either tissue lysates [4,110,112,113,114,115] or serum [116,117,118,119].

Most of these studies have focused on differential protein profiles between TH subtypes or between normal thymus and THs and have typically found significant differences based on hundreds of differentially expressed proteins, some of which have also been proposed in a diagnostic context for refined TH classification [110]. Reverse phase protein array (RPPA) data from the TCGA dataset were used to identify significant pathways with differential expression between TH subtypes and identified cell cycle, apoptosis, EMT, RAS/MAPK, hormone signaling, breast reactive, and core reactive pathway [4]. Altered receptor tyrosine kinase (RTK) signaling in THs has been the focus of considerable interest over the past decades but has often been based on small series using immunohistochemical techniques or RT-PCR. Many of these studies have identified significant differences between TH subtypes, and the overexpression of RTKs was often associated with more aggressive behavior or advanced tumor stage. The best-studied examples are the EGFR (positive in 43–100% of THs) [120,121,122,123,124], IGF1R (positive in 43–100% of THs) [125,126,127,128], VEGFR 1-3 [129,130,131], and their ligands VEGFA, VEGFC, and VEGFD [131]. THs are consistently negative for KIT [132,133,134], ALK [135], HER2 [136,137], and MET [137]. However, the overexpression of a protein does not necessarily prove its functional relevance (e.g., as a predictive biomarker for therapeutic response). Using phospho-RTK arrays, Küffer et al., studied the activation of 44 RTKs in 37 type B2 and B3 thymomas [138]. EGFR was by far the most prevalent active RTK (74% of cases). However, there was a striking dichotomy between primary and metastatic tumors: all metastatic THs were negative for EGFR, and 57% of these tumors instead showed TYRO3/Dtk activation. The most frequently activated other RTKs in THs were (in descending order) FGFR2 (30%), VEGFR3 (24%), TRKC (24%), Tie2, TRKB and EphA6 (21%), VEGFR1 (19%), EphB6, FGFR4, FLT3, INSR, VEGFR2 (16% each), and ERBB3. Interestingly, this study also showed the activation of RTKs that are not normally overexpressed in TH, namely ERBB2 (11%) and KIT (8%). Similarly, a recent study [139] described KITLG overexpression preferentially in type A and AB thymomas, which are consistently KIT negative by immunohistochemistry. KITLG overexpression was associated with highly specific changes in the mRNA and miRNA expression profiles, the upregulation of DNMT3B and downregulation of DNMT1, DNMT3A, and DNMT3L as well as altered methylation patterns. The major upregulated pathway in THs upon KITLG overexpression was MAPK signaling.

## 8. Conclusions and Future Directions

Despite their heterogeneity and the lack of driver mutations in most THs, key principles and molecular features are emerging that may help to improve our understanding of TH biology and open up opportunities for targeted treatments. The TCGA data have shown that the heterogeneity of THs at the molecular level can be reduced to two main categories, namely type A and AB thymomas and type B thymomas. While it may still be relevant to investigate the exact mechanisms that differ, for example, between type B1 and B2 thymomas, the main research effort should probably focus on elucidating the molecular pathogenesis of the two main subgroups. Type A/AB thymomas carry no or few chromosomal alterations but a recurrent GTF2I mutation as the basic driver. Type B thymomas harbor a variable number of chromosomal alterations but no known driver mutations. In both settings, additional non-mutational mechanisms shape the behavior of the cell by altering critical “hallmark” pathways such as cell cycle regulation or resistance to apoptosis (Table 1).

## Figures and Tables

**Figure 1 cancers-16-00942-f001:**
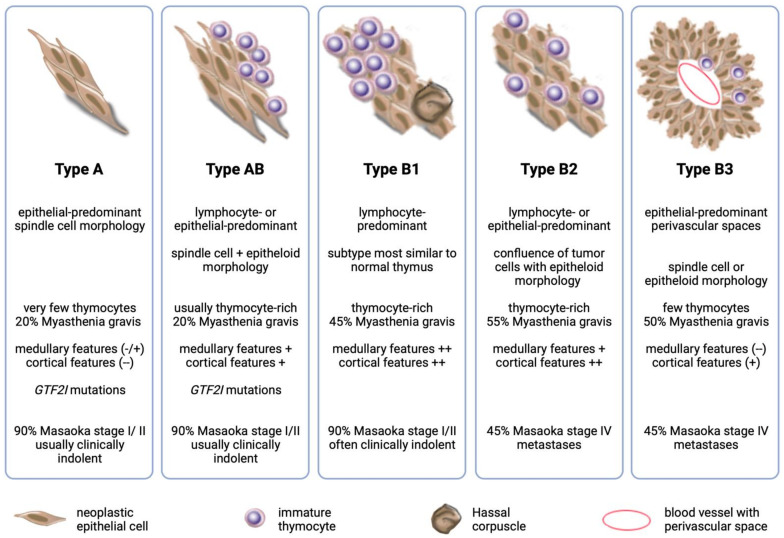
Summary of important histological and clinical features of the main thymoma subtypes. (− − feature not pesent, −/+ feature absent or only partially present, + feature present, ++ feature fully developed compared to normal thymus).

**Figure 2 cancers-16-00942-f002:**
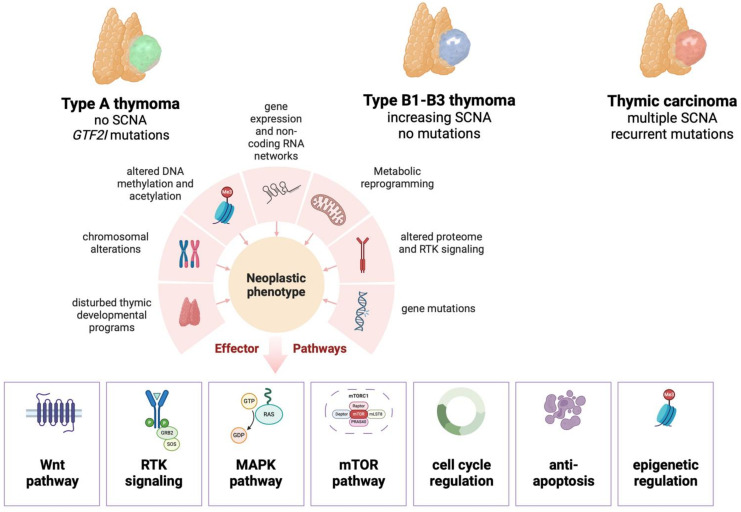
Non-mutational mechanisms and “hallmark” molecular pathways in thymomas. Perturbations at different levels of cellular control converge into the deregulation of critical signaling pathways. SCNA: somatic copy number alterations; RTK: receptor tyrosine kinase.

**Table 1 cancers-16-00942-t001:** Summary of key molecular findings related to the “hallmark” altered signaling pathways in thymomas.

Pathway	Summary	References
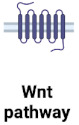	Wnt signaling plays an important role in the development and function of the normal thymus (e.g., through the activation of FOXN1); Wnt inhibitors induce TEC senescence and thymic involution; THs show increased non-canonical Wnt signaling; Altered gene expression of Wnt signaling genes in THs; Wnt signaling is increased in THs with *GTF2I* mutations.	[14,17,18,19] [17,18,19] [21,22] [78,82] [82]
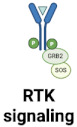	IGF and FGF signaling is critical for the development of normal TECs; *GTF2I* mutation increases receptor tyrosine kinase signaling; Overexpression of EGFR, IGF1R, and VEGFR observed in a high percentage of THs; Activation of pEGFR or pTYRO3/DTK occurs in 85% of type B2 and B3 THs; Frequent overexpression of KITLG in type A and AB THs.	[13,14,15,16] [4] [121,122,123,124,125,126,127,128,129,130,131,132] [138] [139]
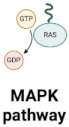	Downregulation of the glycolipid pathway enzyme ST3GAL6 activates MAPK signaling and predicts poor prognosis in THs; MAPK signaling was one of the main signaling pathways with significant differences between TH subtypes in the TCGA dataset; Overexpression of KITLG in type A and AB THs leads to activation of MAPK signaling.	[105] [4] [139]
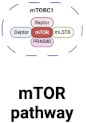	Overexpression of the chr19q13.42 (C19MC) cluster in type A and AB THs activates the mTOR pathway; *GTF2I* mutation leads to downregulation of mTOR signaling; mTOR inhibitors reduced cell viability in a type AB TH cell line;	[77] [4] [77]
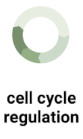	*CDKN2A* mutations found in 6% of THs; alternative mechanisms of *CDKN2A* inactivation through promoter methylation or miR-deregulation likely even more frequent; Several studies described altered expression of cell cycle regulatory genes; *GTF2I* mutation leads to alterations in cell cycle regulation with enrichment of MYC, E2F, and G2M checkpoint target genes; Cell cycle regulation was one of the top pathways associated with metastasis in THs.	[39,40,62,63] [4,72,79,81] [8,31,32] [72]
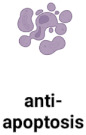	Overexpression or CN gains of anti-apoptotic proteins BCL2, BIRC3, cFLIP, MCL1, and BCL-XL, and downregulation of pro-apoptotic NOXA are a common finding in malignant TETs; BH3 profiling has shown functional dependence of B2 and B3 THs on MCL-1 and BCL-XL; The lncRNA MALAT1 is frequently overexpressed in TETs and mediates apoptosis resistance; Up to 13% of THs show mutations of TP53.	[36,47,48,49,50,51,52] [36,53] [100,101] [29]
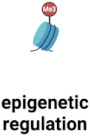	Global methylation studies found significant differences between TH subtypes; Increased DNMT activity and overexpression of class I (HDAC1-3 and 8) and class II (HDAC4-7 and 9-10) HDACs are associated with poor prognosis in THs; The HDAC inhibitor belinostat has shown activity in THs in clinical trials.	[4,63,64,65] [60,68] [70,71]

THs: thymomas, TECs: thymic epithelial cells, RTK: receptor tyrosine kinase, miR: microRNA, TETs: thymic epithelial tumors.

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
