# Peer review of "Non-Mutational Key Features in the Biology of Thymomas"

_cancers, 2024, doi:10.3390/cancers16050942_

Round 1

Reviewer 1 Report

Comments and Suggestions for Authors

 I had the pleasure to review this interesting manuscript by Kuffer and colleagues entitled “Non-mutational key features in the biology of thymomas”.

 Herein, authors summarized most recent evidence on common non-mutational key features such as chromosomal, epigenetic, and metabolic alterations in thymomas.

Authors made a big effort to sketch out and summarize alle the evidence available in literature on the molecular features of thymomas despite the lack of driver mutations (except for GTF2I in Type A/AB thymomas).

I found very useful grouping all the molecular traits according to the role of the pathway and the kind of molecular and genetic features.

I have not found any major issues or concerns and I believe that the manuscript could deserve a publication on such prestigious journal, but I have just some small tips.

-          Since this manuscript represents a narrative review, authors should briefly describe the articles selection criteria.

-          Each paragraph could benefit by a small table to point out the main topic/features of the following text.

-          A comprehensive table of the main molecular features recognized or better evaluated could help the reader understand the text.

-          Discussione could benefit from the reading of the following article doi:10.3389/fonc.2023.1224491. PMID: 37671056; PMCID: PMC10475716.

Author Response

We want to thank the reviewer for the constructive feedback.

We have addressed all issues raised (changes in the text marked in yellow):

...authors should briefly describe the articles selection criteria

a short respective methodological sentence was added in line 75 f.

-          Each paragraph could benefit by a small table to point out the main topic/features of the following text.

since this would have disturbed the flow of the manuscript and since we have added the comprehensive table requested below, we would prefer to omit this addition

-          A comprehensive table of the main molecular features recognized or better evaluated could help the reader understand the text.

A new detailed table 1 with the requested features was added to the manuscript

-          Discussione could benefit from the reading of the following article doi:10.3389/fonc.2023.1224491. PMID: 37671056; PMCID: PMC10475716.

The new citation (no 29 in the manuscript) was added and mentioned in two paragraphs

Reviewer 2 Report

Comments and Suggestions for Authors

In this review article, the authors tried to integrate the existing knowledge of thymomas (TH) biology into a broader picture and highlight opportunities for targeted treatment options in these non-mutated tumors.

Comments:

This is an interesting review article. The manuscript is well-written. The reviewer has only some minor concerns as follows:

1.     In Figure 1, please correct the typing errors for “epithel-predominant” (epithelial-predominant). Moreover, in the image for Type B3, what is the mean for red circle? It can be indicated in the image or figure legend.

2.     In Figure 2, the whole names for abbreviations can be shown in the figure legend.

3.     In line 220, the “TMs” is mentioned. What is the difference between TH and TMs? It can be explained.

4.     In line 231, please correct the typing error for “HDCA inhibitor” (HDAC inhibitor).

Author Response

We want to thank the reviewer for the careful reading and the constructive feedback.

We have addressed all issues raised (changes in the text marked in yellow):

In Figure 1, please correct the typing errors for “epithel-predominant” (epithelial-predominant). Moreover, in the image for Type B3, what is the mean for red circle? It can be indicated in the image or figure legend.

The figure was changed accordingly, and we have added a graphical legend explaining e.g. the red circle. 

In Figure 2, the whole names for abbreviations can be shown in the figure legend.

We have added the whole names for abbreviations.

In line 220, the “TMs” is mentioned. What is the difference between TH and TMs? It can be explained.

Thank you; this was a typo (-we corrected).

In line 231, please correct the typing error for “HDCA inhibitor” (HDAC inhibitor)

Thank you (we corrected).

Reviewer 3 Report

Comments and Suggestions for Authors

The authors have integrated existing knowledge of thymoma biology from the aspects of disturbed thymic developmental programs, functional implications of chromosomal alterations, altered DNA methylation and acetylation, altered gene expression and non-coding RNA networks, metabolic reprogramming, and proteomics and altered tyrosine kinase signaling, providing insights into future research directions. I have some questions as follows:

1.Please verify whether the abbreviation for thymomas is TH.

2.Figure 1 lacks annotations, such as what the purple color represents, and the meaning of the "-/+" signs.

3.TEC and sCNAs lack their full names; only abbreviations are provided. Please double-check.

4.There are inconsistencies in the writing conventions of gene names and protein names. Some places use the gene name format to represent proteins. Please review this carefully.

Author Response

We want to thank the reviewer for the constructive feedback.

We have addressed all issues raised (all changes in the text marked in yellow):

1. Please verify whether the abbreviation for thymomas is TH

Thank you - there were some inconsistencies (we changed to TH throughout)

2. Figure 1 lacks annotations, such as what the purple color represents, and the meaning of the "-/+" signs.

Fig. 1 was extensively revised, and a short graphical legend as well as an explanation for "-/+" etc. was added

3.TEC and sCNAs lack their full names; only abbreviations are provided. Please double-check.

Thank you, both full names were now added upon first apearance

4. There are inconsistencies in the writing conventions of gene names and protein names. Some places use the gene name format to represent proteins. Please review this carefully.

Thank you, we checked and corrected throughout.